# Enzyme-Mediated Amplification (EMA) for Detection of the Pinewood Nematode *Bursaphelenchus xylophilus*

**Li-Chao Wang, Min Li** **, Ruo-Cheng Sheng and Feng-Mao Chen** *

Collaborative Innovation Center of Sustainable Forestry in Southern China, College of Forestry,
Nanjing Forestry University, Nanjing 210037, China
* Correspondence: cfengmao@njfu.edu.cn

**Abstract:** The pinewood nematode (PWN), *Bursaphelenchus xylophilus*, is a notorious parasitic nematode of pine trees that causes pine wilt disease (PWD), leading to extensive mortality of different pine species around the world and considerable economic losses, thus posing a threat to healthy pines worldwide. Fast and accurate detection technology is necessary for the management of PWD spread. This study describes the development of a new DNA extraction method and detection technology, enzyme-mediated amplification (EMA), using primers and a newly designed probe according to the rRNA internal transcribed spacer gene ITS2. The detection process can be completed within 40 min, including DNA extraction for 10 min and detection for 30 min, by exploiting the synergistic action of multiple enzymes. This method can detect PWNs from different geographic areas quickly and accurately at all life stages, singly or in a mixture, and can distinguish PWNs from other species of the *Bursaphelenchus* group, showing that it is not only reliable but also rapid, greatly improving the efficiency and speed of PWN detection. Therefore, the technology is expected to be highly beneficial in PWN quarantine testing.

**Keywords:** detection technology; pine wilt disease; enzyme-mediated amplification; quarantine

## 1. Introduction

Pine wilt disease (PWD), an incurable disease of pine trees that leads to occlusion of the xylem, is caused by the pinewood nematode (PWN) *Bursaphelenchus xylophilus* [(Steiner & Buhrer, 1934) Nickle, 1937] which is a global quarantine microorganism [1]. The disease is endemic to the United States, where exotic, nonnative pine species are the main target, but has spread to cause serious damage to pine trees in Asia (China, Japan and Korea) and Europe (Portugal and Spain). The PWN was first detected in China on black pine (*Pinus thunbergii*) in Nanjing city, Jiangsu Province, China [2], but has since spread to 666 counties in 18 provinces of China through the timber trade, human activities and vector beetles, despite taking many measures to slow its spread [3]. Much research has been carried out to protect pine trees and to reduce economic losses, including on the nematode, host trees, vector beetles (pine sawyers, *Monochamus* spp.) and the environment [4–10]. However, there are still no effective measures with which to control the occurrence of PWD, such that strengthening quarantine protocols is still important in PWD management.

There are a number of plant–parasitic nematode species of the genus *Bursaphelenchus* found on pine [11]. The morphological characteristics of the PWN, including its juvenile and adult male characteristics, are similar to those of other species of the *xylophilus* group. Adult females of the PWN are the only group that can be used to distinguish the species from other *Bursaphelenchus* spp. because they have distinctive morphological characteristics, including details of the vulval flap, stylet, tail shape, etc. [12]. Special situations may further limit the use of morphological traits to distinguish *Bursaphelenchus* spp. The PWN has two life forms: propagative and dispersive [13]. When the environment is not suitable for the survival of PWNs (e.g., low temperature, drought or a lack of food), the PWN will change form from the

propagative type, with typical morphological characteristics, to the dispersive type, with loss of certain morphological characteristics, including disappearance of the median bulb and the stylet [14,15], such that traditional identification using morphological characters of adult females cannot be adopted to identify the species. In addition, *B. xylophilus* extracted from wood samples have mucronate tails [16], which are very similar in morphology to those of *Bursaphelenchus mucronatus* Mamiya & Enda. Therefore, traditional morphological characteristics cannot be used to accurately distinguish between species.

The PWN is a serious, invasive, plant–parasitic nematode that is believed to have originated in North America, where it is considered to be endemic. However, Japan was the first country outside the native range to report the occurrence of PWD in 1905, and Japanese pines (such as *Pinus thunbergii* Parl. and *Pinus densiflora* Siebold & Zucc.) are regarded as being highly susceptible to PWD. A large-scale outbreak of PWD in Japan resulted in extensive mortality of native pine trees. Mamiya proposed that PWD cannot occur in areas where the annual mean temperature is lower than 10 °C, based on the relationship between disease development and local temperature [17]. However, PWNs have successfully adapted to temperatures below 10 °C and spread to Liaoning Province (annual average temperature < 10 °C) [18], explaining PWD diffusion from warm areas to cold areas. Intraspecific variation in the PWN occurs in different geographical environments [19]. To date, the PWN has been detected in 18 provinces of China, covering four climatic zones, namely, tropical, subtropical, temperate and middle temperate zones.

In recent years, increasing numbers of molecular techniques have been developed to overcome the shortcomings of traditional morphology-based identification technologies. Most PWN identification techniques are based on polymerase chain reaction (PCR), including nested PCR and real-time PCR [20–23], although others do not involve PCR, such as loop-mediated isothermal amplification (LAMP) [24,25], recombinase polymerase amplification (RPA) [26], and denaturation bubble-mediated strand exchange amplification (SEA) [27]. The results of these molecular detection systems can be read by the naked eye, in contrast to the tedious and technical approach of morphological trait measurement, which must be performed by highly trained professionals to detect such small differences.

Thus, there is a demand for a sensitive, accurate and rapid test, particularly for quarantine testing purposes. To this end, we developed an enzyme-mediated amplification (EMA) test to detect the PWN to overcome the current limitations of PWN identification tests. This amplification technique can shorten the amplification time in two ways, namely, avoiding the need for heating and cooling the sample, compared with conventional PCR amplification experiments, and amplifying the target gene by using multiple enzymes at the same time (Figure 1), with the latter being the main reason for the shortened test time. In addition, EMA is easy to perform, obviating the need for professional knowledge or technical experience. Thus, this detection method is expected to provide a fast and accurate strategy for all PWN testers, including nonexperts, of particular significance in quarantine situations.

## 2. Materials and Methods

### 2.1. Nematodes

The 24 nematode strains tested consisted of 12 isolates of *B. xylophilus*, 10 isolates of *B. mucronatus*, and 1 strain each of *Bursaphelenchus sinensis* and *Bursaphelenchus vallesianus* Braasch, Schonfeld, Polomski, et al., 2004, all of which were preserved in the Jiangsu Key Laboratory for Prevention and Management of Invasive Species, Nanjing Forestry University. Except for the Bm10 strain of *B. mucronatus* (from Japan), the strains were isolated from diseased wood samples from different sites in China (Table 1).

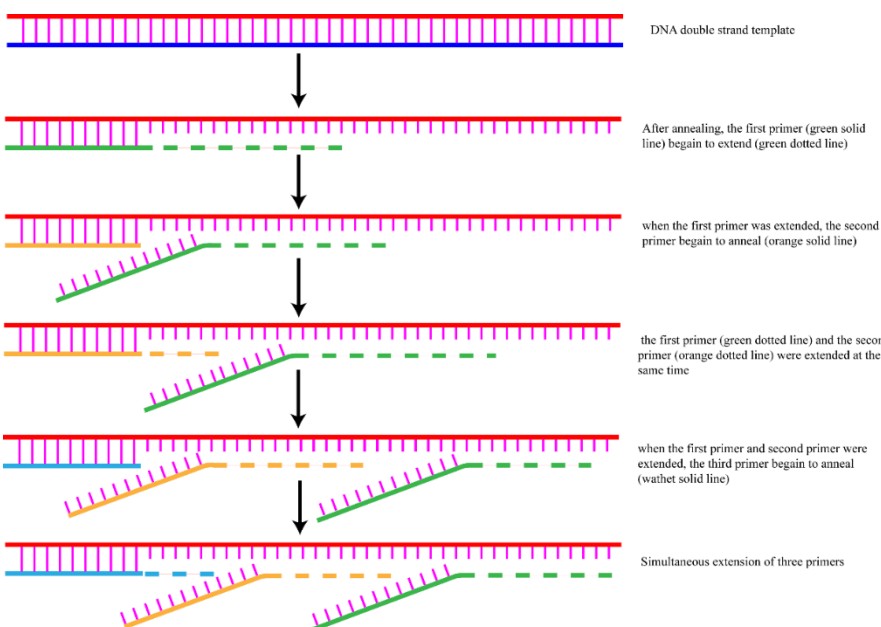

**Figure 1.** Schematic illustration of enzyme-mediated amplification.

**Table 1.** Geographic origins and hosts of *Bursaphelenchus* species used in the study.

| Species | Strain Name | Geographical Origin | Host Species |
|---|---|---|---|
| *B. xylophilus* | Bx01 | Huangshan, Anhui Province | *P. massoniana* |
| | Bx02 | Nanjing, Jiangsu Province | *P. thunbergii* |
| | Bx03 | Fuyang, Zhejiang Province | *P. massoniana* |
| | Bx04 | Yueyang, Hunan Province | *P. massoniana* |
| | Bx05 | Jingmen, Hubei Province | *P. massoniana* |
| | Bx06 | Qingdao, Shandong Province | *P. thunbergii* |
| | Bx07 | Dalian, Liaoning Province | *P. thunbergii* |
| | Bx08 | Shanyang, Shaanxi Province | *P. thunbergii* |
| | Bx09 | Pengze, Jangxi Province | *P. massoniana* |
| | Bx10 | Renhuai, Guizhou Province | *P. massoniana* |
| | Bx11 | Quanzhou, Fujian Province | *P. massoniana* |
| | Bx12 | Xinyang, Henan Province | *P. massoniana* |
| *B. mucronatus* | Bm01 | Nanling, Anhui Province | *P. massoniana* |
| | Bm02 | Nanling, Anhui Province | *P. massoniana* |
| | Bm03 | Huangshan, Anhui Province | *P. massoniana* |
| | Bm04 | Yichang, Hubei Province | *P. massoniana* |
| | Bm05 | Pingnan, Fujian Province | *P. elliotii* |
| | Bm06 | Fumin, Yunnan Province | *P. massoniana* |
| | Bm07 | Fushun, Sichuan Province | *P. elliotii* |
| | Bm08 | Fushun, Sichuan Province | *P. elliotii* |
| | Bm09 | Huizhou, Guangdong Province | *P. massoniana* |
| | Bm10 | Japan | *P. thunbergii* |
| *B. sinensis* | Bs01 | Pengze, Jangxi Province | *P. massoniana* |
| *B. vallesianus* | Bv01 | Nanjing, Jiangsu Province | *P. massoniana* |

## 2.2. DNA Extraction

DNA was extracted from mixed life-stage samples of nematodes: 20 μL of nematodes (about 30 individuals) were placed into a 1.5 mL tube filled with 20 μL of lysis buffer (0.8 mM Tris-HAc, 0.08 mM EDTA, 0.02% SDS, 0.08% NP-40) and then vigorously mixed and incubated at 95 °C for 10 min in a Chb-t1 constant-temperature heater (Shanghai Woyuan Technology Co., Ltd., Shanghai, China). Then, the tube was centrifuged at 12,000× *g* for 3 min, and 3.5 μL of the supernatant (containing DNA) was used for EMA.

DNA was also extracted from individual nematodes, including 2nd-stage juveniles (J2), 3rd-stage juveniles (J3), 4th-stage juveniles (J4) and adults. Individual nematodes were selected using a pipette and transferred into 5 μL of lysis buffer under a DM500 microscope (Leica, Germany). The remaining DNA extraction steps were performed as described above.

### 2.3. Primer and Probe Design

The specific primers and probe were designed after aligning the ITS2 sequences from *B. xylophilus* (GenBank accession number: KM657966.1) and *B. mucronatus* (GenBank accession number: JF826237.1) by using Oligo 7 Primer Analysis Software [28] (Figure 2). The forward primer sequence was 5'-TTAAACTCGAGCAGAAACGCCGACTTG-3', and the reverse primer sequence was 5'-CTCCAAACATTCTCATCCGAACGTCCCT-3'. The probe was designed as follows: sequence, 5'-CCCTCTCGCCCCGCACGGACAAACAG/i6FAMd T/G/idSp/G/iBHQ1dT/AGAAGATATTGGTC-3' C3 spacer; i6-FAMdT, 6-carboxyfluorescein-labeled DT nucleotides; idSp, base loss; iBHQ, DT nucleotides labeled with Black Hole Quencher (BHQ) quenched group; and C3, 3'-hydroxyl sealing. The primer pairs and the probe were synthesized by Suzhou Click Gene BioTech Co., Ltd. (Jiangsu, Suzhou, China).

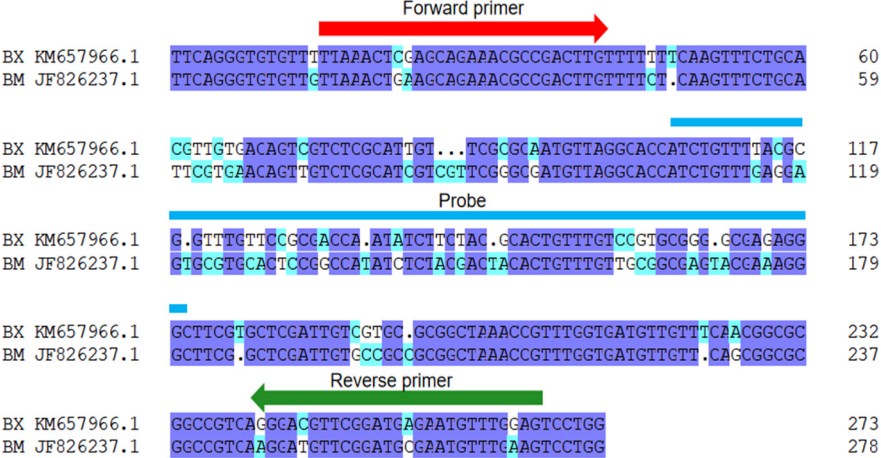

**Figure 2.** Nucleotide sequence alignment of the ITS2 genes from *B. xylophilus* (BX) and *B. mucronatus* (BM). The locations of the probe and forward and reverse primers are marked by blue, red and green boxes, respectively.

### 2.4. EMA

The EMA reaction system consists of two parts: amplification and fluorescence signal detection. The amplification process includes helicase, DNA polymerase, single strand DNA-binding (SSB) protein and ATP regeneration protein. The role of the helicase is to open the double chain structure of the template nucleic acid. The single-chain binding protein can form a complex with the primers, which can recognize and bind to the target sequence specifically. Then, the DNA polymerase continues to extend the DNA chain to achieve amplification of the target fragment. The ATP regeneration protein can provide energy for the whole reaction system. Detection of the fluorescence signal requires an endonuclease. The probe is complementary to the target amplification fragment, and the endonuclease in the detection system can recognize the base deletion site, hydrolyze the probe and release the fluorescence signal.

For the strand exchange amplification (SEA) reaction, 10 μL of complex solution and 10 μL of DNA template were put into a 0.2 mL tube containing 0.5 g of freeze-dried powder containing 100 μM forward primer, 100 μM reverse primer, 10 μM probe, 300–600 ng/μL DNA helicase, 1500–2500 ng/μL SSB protein, 300–600 ng/μL DNA polymerase, 20–50 ng/μL ATP regeneration protein, 300–600 ng/μL DNA restriction endonuclease, 1000–2000 ng/μL RNase inhibitor, 1 M creatine phosphate, disodium salt, 1–2 M Tris-Ac (pH = 8.0) and 25 mM dNTP. The tube contents were vigorously mixed and then incubated at 42 °C for 30 min in a Click i detection instrument (Suzhou ClickGene, China), which collects fluorescence data every 30 s and presents the time-fluorescence signal graph. The CT value indicates the number of cycles in which the fluorescence signal in each reaction tube reached the set threshold.

### 2.5. Specificity and Sensitivity of the SEA Reaction

The specificity of the SEA reaction was evaluated by detecting the DNA template extracted from nematode strains listed in Table 1. The sensitivity of the SEA reaction was assessed by detecting the DNA template extracted from single nematodes. When the amplification curve of the sample is "S-shaped", the result is considered positive; otherwise, it is considered negative.

## 3. Results

### 3.1. Detection of PWNs from Different Geographical Sources

The detection results and amplification curves of 12 DNA samples extracted from *Bursaphelenchus* from 12 provinces are shown in Figures 3 and 4. The detection results from the 12 DNA samples were consistent with that of the positive control, and the amplification curves of the DNA samples were similar to that of the positive control, indicating that the EMA method can accurately detect PWNs from different geographical sources.

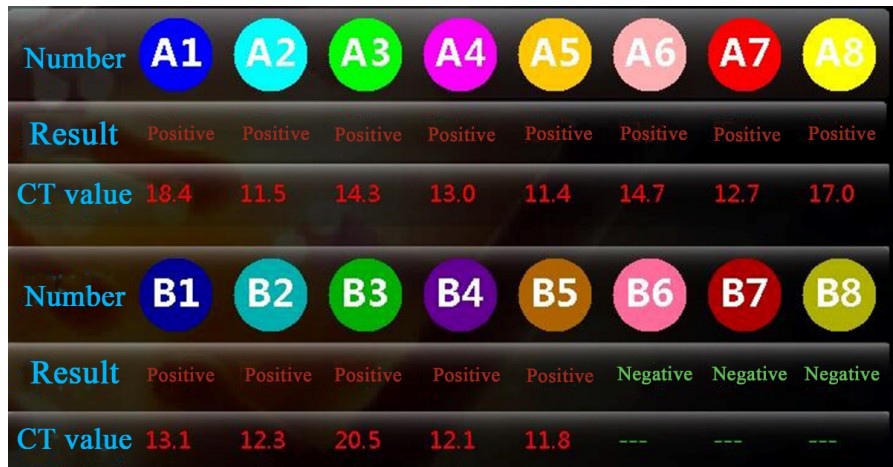

**Figure 3.** The results of DNA detection of PWNs from different geographical sources by EMA.A1–A8 and B1–B6 are the well numbers. The corresponding test samples of wells A1–A8 are Bx01–Bx08, and the corresponding test samples of wells B1–B4 are Bx09–Bx12; well B5 is the positive control, and well B6 is the negative control. B7 and B8 are empty slots.

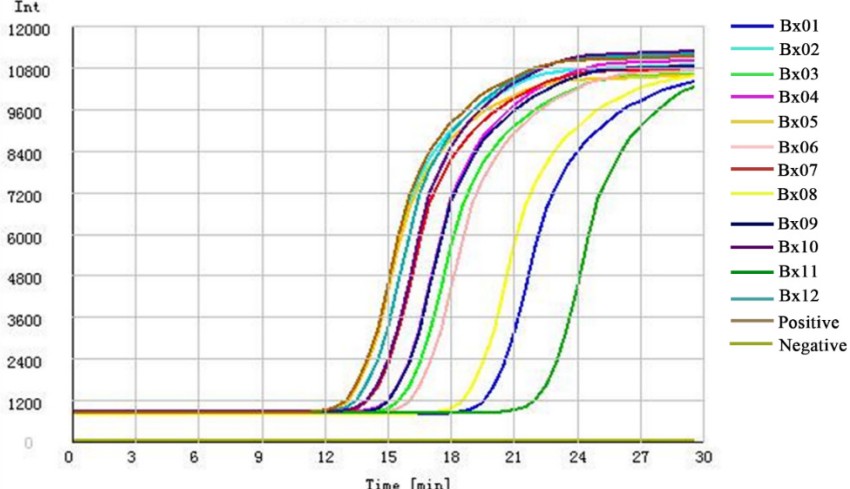

**Figure 4.** DNA amplification curve resulting from EMA of *B. xylophilus* PWNs from different geographical sources.

### 3.2. Detection of Different Nematode Strains and Species

The detection results and amplification curves for DNA samples from different nematode strains and species are shown in Figures 5 and 6. Of the 14 samples tested, only the positive control showed the S-type amplification curve (Figure 5). The amplification results of *B. mucronatus* from 10 geographical sources and single-nematode samples of *B. sinensis* and *B. vallesianus* were consistent with those of the negative control (Figure 5); in addition, they exhibited similar amplification curves. These results show that EMA can accurately distinguish PWNs from different, closely related nematodes.

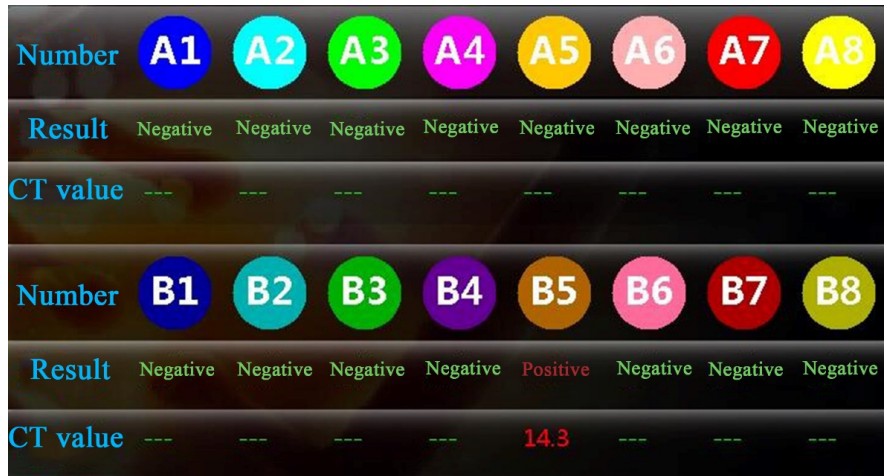

**Figure 5.** The results of detection of different nematodes by EMA.A1–A8 and B1–B6 are the well numbers, and the corresponding test samples of wells A1–A8 are Bm01–Bm08. The corresponding test samples of wells B1–B4 are Bm09, Bm10, Bs01 and Bv01, respectively; well B5 is the positive control, and well B6 is the negative control. B7 and B8 are empty slots.

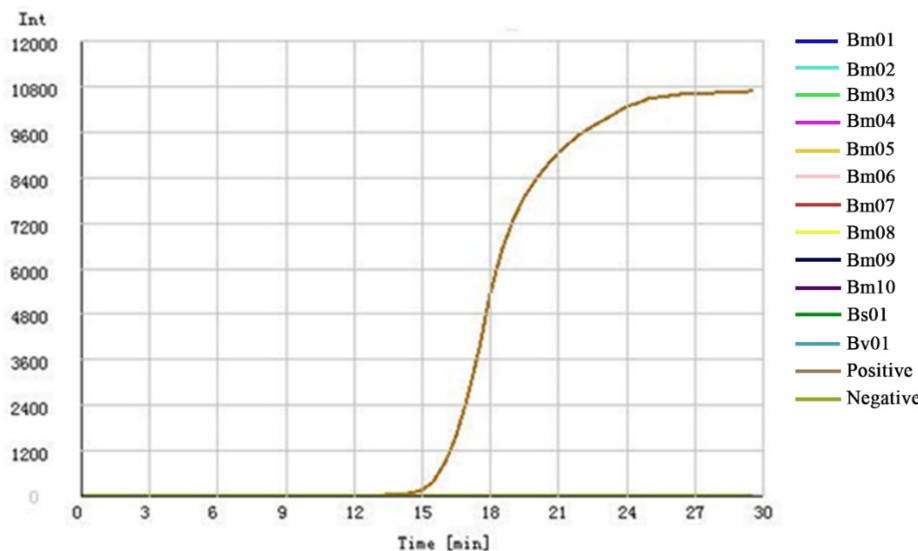

**Figure 6.** DNA amplification curve from EMA of different strains and *Bursaphelenchus* species.

### 3.3. Detection of Individual Nematodes at Different Stages of Development

The detection results and amplification curves of DNA extracted from single nematodes are shown in Figures 7 and 8. The results of DNA detection from single nematodes were consistent with those of the positive control; the amplification curves of DNA extracted from individual nematodes at the $J_2$, $J_3$ and $J_4$ juvenile and adult stages were similar to those of the positive control, indicating that the EMA method was sensitive and could detect individual nematodes.

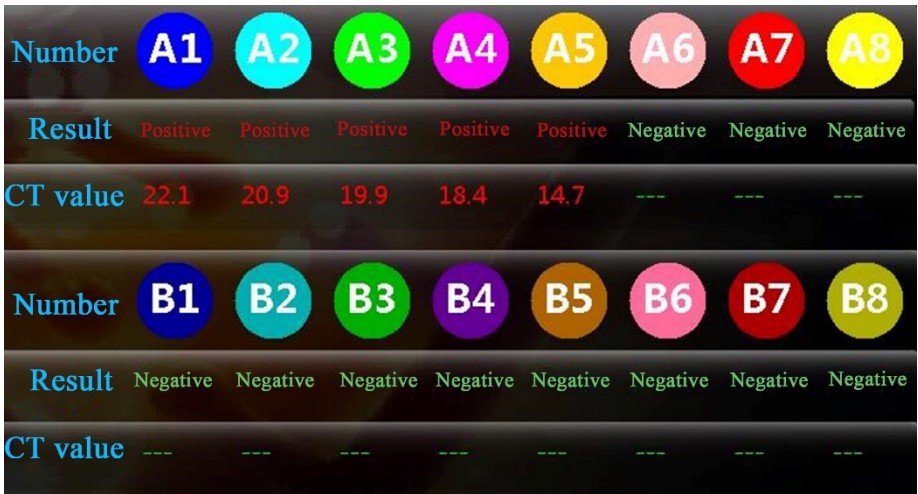

**Figure 7.** The results of DNA detection of single nematodes by EMA. The corresponding test samples of wells A1–A4 are J$_2$, J$_3$, and J$_4$ juveniles and adults, respectively. Sample A5 was the positive control, whereas A6 was the negative control. A7, A8 and B1–B8 are empty slots.

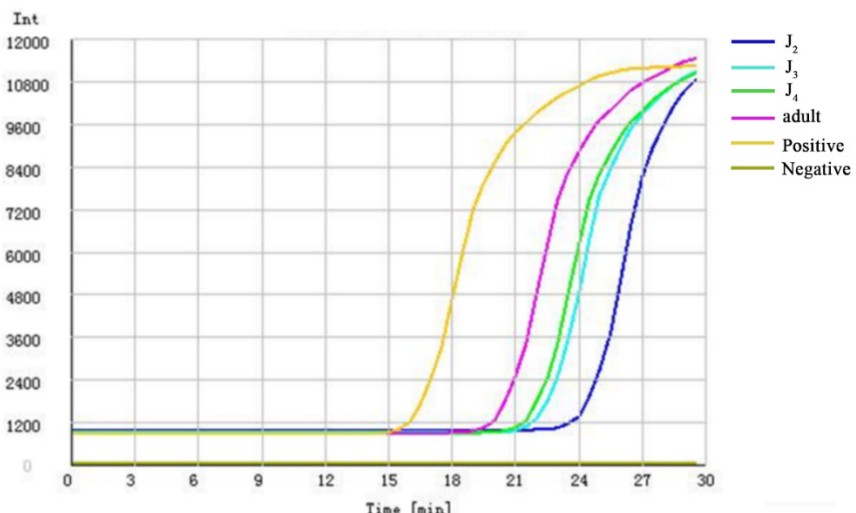

**Figure 8.** The amplification curve of DNA of individual nematodes at different stages of development resulting from EMA.

## 4. Discussion

In this study, we described a novel method (EMA) for detecting PWNs. In the current study, the primers and probe were able to accurately detect all strains of the PWN collected from the 12 provinces. Compared with other detection technologies based on PCR, this method can save a considerable amount of time needed to obtain a result. EMA takes only 40 min to obtain a positive/negative result: 10 min for the DNA extraction step and 30 min for the EMA assay. Meanwhile, the EMA assay can also detect PWNs accurately (at all stages of development, from mixed-stage samples, and for all strains of *B. xylophilus*), avoiding false-positive diagnoses resulting from small differences in morphological variables that cannot be observed qualitatively by the naked eye. EMA can also distinguish *B. xylophilus* from other *Bursaphelenchus* spp.

Among *Bursaphelenchus* species, *B. mucronatus* is considered to be closely related to *B. xylophilus* (the PWN), based on their similar taxonomic positions, life cycles, hosts and even vector beetles [29]. The traditional morphological identification technique relies only on the length of the adult female's tail tip to distinguish the PWN from *B. mucronatus* [15], such that *B. mucronatus* is sometimes misidentified as the PWN because of morphological overlap between the two species. Furthermore, some studies have reported that the two

nematode species can mate and generate viable and reproducible progeny [27], suggesting that they may not be distinct species. We have shown that the primers and probe developed for EMA can distinguish *B. xylophilus* not only from *B. mucronatus* but also from other *Bursaphelenchus* species.

Although juveniles and males of different *Bursaphelenchus* spp. cannot be identified on the basis of morphology [13], sometimes only juveniles or adult males, instead of females, may be present in the extraction liquid in which the wood sample is soaked to achieve extraction. The single-nematode test was repeated to include the $J_2$ to $J_4$ and adult stages to verify the sensitivity of the primers and probe. The EMA detection technology was able to accurately detect juveniles of different ages.

In conclusion, the EMA technique presented here provides a more specific, simpler, more sensitive and more rapid detection method for the PWN compared with conventional PCR or morphological testing. Therefore, this technique is expected to provide an appropriate, rapid diagnostic tool for all users, especially in new areas of PWD occurrence.

**Author Contributions:** Conceptualization, L.-C.W. and F.-M.C.; methodology, F.-M.C.; formal analysis, M.L.; investigation, R.-C.S.; resources, L.-C.W.; data curation, L.-C.W.; writing—original draft preparation, L.-C.W.; writing—review and editing, M.L. All authors have read and agreed to the published version of the manuscript.

**Funding:** The work was supported by the National Key Research and Development Program of China (Grant Number: 2021YFD1400903).

**Institutional Review Board Statement:** Not applicable.

**Informed Consent Statement:** Not applicable.

**Data Availability Statement:** Not applicable.

**Conflicts of Interest:** The authors declare no conflict of interest.

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
