# Peer review of "Enzyme-Mediated Amplification (EMA) for Detection of the Pinewood Nematode Bursaphelenchus xylophilus"

_forests, doi:10.3390/f13091419_

Round 1
Reviewer 1 Report
This study by Wang et al. deals with enzyme-mediated amplification (EMA) for detection of the pinewood nematode Bursaphelenchus xylophilus. I think that this paper will help to improve the efficiency of the diagnosis of pinewood nematode. However, the manuscript needs some extra works to improve the manuscript for the publication. I give some comments below:
- Pinewood nematode is bisexual. So why didn't the author use the mitochondrial gene (COI or COII)? it would be better to add an explanation for the reason.
- Page 3 line 88: What is the average density of nematodes in a sample of 20μL?.
- I recommend adding standard curves according to DNA concentration for reproducibility of experimental methods and results presented in this manuscript.
Author Response
Comments to the Author
1) This study by Wang et al. deals with enzyme-mediated amplification (EMA) for detection of the pinewood nematode Bursaphelenchus xylophilus. I think that this paper will help to improve the efficiency of the diagnosis of pinewood nematode.
Response: Thank you very much for your patient review and comments.
2) Pinewood nematode is bisexual. So why didn't the author use the mitochondrial gene (COI or COII)? it would be better to add an explanation for the reason.
Response: Thank you for your advice, but we think the mitochondrial gene was too common to be used for, so we choose the other gene.
3) Page 3 line 88: What is the average density of nematodes in a sample of 20μL?
Response: Thank you for your advice, the density has been added:
DNA was extracted from mixed life-stage samples of nematodes: 20 μL of nematodes (about 30 individuals) were placed into a 1.5 mL tube filled with 20 μL of lysis buffer (0.8 mM Tris-HAc, 0.08 mM EDTA, 0.02% SDS, 0.08% NP-40) and then vigorously mixed and incubated at 95°C for 10 min in a Chb-t1 constant-temperature heater (Shanghai Woyuan Technology Co., Ltd., China).
4) I recommend adding standard curves according to DNA concentration for reproducibility of experimental methods and results presented in this manuscript.
Response: Thank you for your advice. We are very sorry that because of the COVID-19 we can’t back to college to copy that DNA concentration picture.
Reviewer 2 Report
New detection method for Burspahelenchus species is the main question addressed by the research. Looking to the importance of the nematode, the topic is quite original. A new detection method is always a good contribution. The paper is well written, the text is clear and easy to read.
Every species (scientific name), when reported for the first time in the text should be written in full with Authority and systematics.

Author Response
Comments to the Author
1) Every species (scientific name), when reported for the first time in the text should be written in full with Authority and systematics.
Response: Thanks for your advice, the Authority and systematics have been added.
E.g: Bursaphelenchus xylophilus [(Steiner & Buhrer, 1934) Nickle, 1937]
Bursaphelenchus mucronatus Mamiya & Enda
2) it should be better to insert keywords not present in the title.
Response: Thanks for your advice, the Keywords have been modified.
Keywords: detection technology; pine wilt disease; enzyme-mediated amplification; quarantine
3) about the Bursaphelenchus cycle, better to write details in the first part (introduction). There are also in this final chapter too many repetitions. Discussion/conclusion should be revised.
Response: Thank you for your advice, the Introduction/Discussion/Conclusion has been revised.
e.g. In this study, we described a novel method (EMA) for detecting PWNs. In the current study, the primers and probe were able to accurately detect all strains of the PWN col-lected from the 12 provinces. Compared with other detection technologies based on PCR, this method can save a considerable amount of time needed to obtain a result. EMA takes only 40 min to obtain a positive/negative result: 10 min for the DNA ex-traction step and 30 min for the EMA assay. Meanwhile, the EMA assay can also de-tect PWNs accurately (at all stages of development, from mixed-stage samples, and for all strains of B. xylophilus), avoiding false-positive diagnoses resulting from small differences in morphological variables that cannot be observed qualitatively by the naked eye. EMA can also distinguish B. xylophilus from other Bursaphelenchus spp., which are not associated with PWD.
Among Bursaphelenchus species, B. mucronatus is considered to be closely related to B. xylophilus (the PWN), based on their similar taxonomic positions, life cycles, hosts, and even vector beetles[29]. The traditional morphological identification technique relies only on the length of the adult female’s tail tip to distinguish the PWN from B. mucronatus [15], such that B. mucronatus is sometimes misidentified as the PWN because of morphological overlap between the two species. Furthermore, some studies have reported that the two nematode species can mate and generate viable and reproducible progeny[27], suggesting that they may not be distinct species. We have shown that the primers and probe developed for EMA can distinguish B. xylophilus not only from B. mucronatus but also from other Bursaphelenchus species.
Although juveniles and males of different Bursaphelenchus spp. cannot be identified on the basis of morphology[13], sometimes only juveniles or adult males, instead of females, may be present in the extraction liquid in which the wood sample is soaked to achieve extraction. The single-nematode test was repeated to include the J2 to J4 and adult stages to verify the sensitivity of the primers and probe. The EMA detection technology was able to accurately detect juveniles of different ages.
In conclusion, the EMA technique presented here provides a more specific, simpler, more sensitive and more rapid detection method for the PWN compared with conventional PCR or morphological testing. Therefore, this technique is expected to provide an appropriate, rapid diagnostic tool for all users, especially in new areas of PWD occurrence.
Reviewer 3 Report
According to my opinion the article provides new powerful and valuable tool for B. xylophilus diagnostics. I was not able to detect any major or minor issues which would prevent its publication, I think it can be in present form which is quite straightforward and understable.
Author Response
1) According to my opinion the article provides new powerful and valuable tool for B. xylophilus diagnostics. I was not able to detect any major or minor issues which would prevent its publication, I think it can be in present form which is quite straightforward and understable.
Response: Thank you very much for your patient review and comments.
Reviewer 4 Report
The authors propose an Enzyme-mediated amplification (EMA) which is a kind of Nicking enzyme-assisted amplification (NEAA). I support this paper because it is clear that this approach avoid false-positive diagnoses and save time, reducing mistakes in addition. The EMA they propose can distinguish B. xylophilus from other Bursaphelenchus spp. My recommendation is to publish it. Scientifically and methodologically performed according canonical approach.
Perhaps be necessary to clarify the legends regarding Fig.3, Fig. 5 and Fig.7.
Fig.3 and Fig.5: well B6 is the negative control, but also Wells B7 and B8?.
Fig.7:The same happens for Wells A7 and A8. Regarding Wells B1 to B8; why not remove them or explain they are empties not negative?
Author Response
Comments to the Author
1) The authors propose an Enzyme-mediated amplification (EMA) which is a kind of Nicking enzyme-assisted amplification (NEAA). I support this paper because it is clear that this approach avoid false-positive diagnoses and save time, reducing mistakes in addition. The EMA they propose can distinguish B. xylophilus from other Bursaphelenchus spp. My recommendation is to publish it. Scientifically and methodologically performed according canonical approach.
Response: Thank you very much for your patient review and comments.
2) Perhaps be necessary to clarify the legends regarding Fig.3, Fig. 5 and Fig.7.
Response: Thanks for your advice, these Figs’ explanations have been modified.
E.g Figure 7. The results of DNA detection of single nematodes by EMA. The corresponding test samples of wells A1–A4 are J2, J3, and J4 juveniles and adults, respectively. Sample A5 was the positive control, whereas A6 was the negative control. A7- A8, B1 and B8 are empty slots.
3) Fig.3 and Fig.5: well B6 is the negative control, but also Wells B7 and B8?
Response: Thank you for your advice, these Figs’ explanations have been modified.
E.g Figure 7. The results of DNA detection of single nematodes by EMA. The corresponding test samples of wells A1–A4 are J2, J3, and J4 juveniles and adults, respectively. Sample A5 was the positive control, whereas A6 was the negative control. A7- A8, B1 and B8 are empty slots.
4) Fig.7: The same happens for Wells A7 and A8. Regarding Wells B1 to B8; why not remove them or explain they are empties not negative?
Response: In order to ensure the authenticity and consistency of the picture, we intercepted the real-time display of the testing instrument, so A7-A8 and B1-B8 were not erased.